# A DIDE-FPP Composite Algorithm for Accurate Tunnel Fire Location

Hong Jiang [1,*] , Yihan Zhao , Chenyang Wang and Lina Cui

School of Electrical and Electronic Engineering, Changchun University of Technology, Changchun 130012, China
* Correspondence: jianghong@ccut.edu.cn

**Abstract:** We propose a DIDE-FPP composite algorithm to improve the spatial location accuracy of tunnel fires based on the distributed individuals differential evolution (DIDE) algorithm and the four-point positioning (FPP) method. Using the DIDE algorithm to solve the multimodal optimization problems in tunnel fire location can locate more peaks and improve peak finding, and the FPP method is used to process the peak points located using the DIDE and achieve the spatial location which cannot be achieved otherwise using the DIDE method only. We used 20 multimodal test functions to evaluate the performance of the DIDE-FPP algorithm in peak finding and solving MMOPs. Through experimental comparisons with 13 other existing advanced methods, the comprehensive performance of the proposed DIDE-FPP composite algorithm shows advantages to some extents. Additionally, the combined value of PR (peak ratio) and SR (success rate) on up to 20 experimental functions is relatively high. The spatial positioning accuracy of a tunnel fire warning system using this positioning method can reach the centimeter level.

**Keywords:** FBG; spatial positioning; peak finding algorithm; peak ratio; success rate





## 1. Introduction

Tunnels play an important role in the subway transportation network, as do the cross-mountain and cross-sea tunnels in other major transportation systems. The fire safety problem of tunnels, especially those of subways, needs to be solved urgently. Subway tunnels are characterized by being closed, long, and narrow, and with low visibility. Once a fire occurs, the consequences can be extremely dangerous. Most electrical equipment in subway trains works in a high-voltage environment for a long time, and the heat dissipation conditions of underground tunnels are poor, which can easily cause equipment aging and heating, as well as temperature rising, which may lead to fire accidents. Research on the development of a real-time early fire warning system to detect fire hazards in subway tunnels focusing on accurate identification in the early stage of a fire and the capability of taking timely measures to prevent the fire from spreading has great practical significance for urban rail transit safety [1]. Due to the special tunnel environment, traditional smoke and light sensors have limitations. Grating fiber has a strong anti-interference ability, and can realize passive monitoring, fast transmission speed, and fixed-point and accurate monitoring of the whole tunnel. It can continuously and uninterruptedly feedback the temperature of the detection points and monitor the fire occurrence well. Thus, the temperature-sensitive fiber grating sensor is very suitable for monitoring tunnel fire temperature with strong anti-electromagnetic interference, good long-term stability, high measurement sensitivity, and long transmission distance [2].

In 1996, Kersey A.D. demonstrated that a distributed temperature sensor system can operate with a temperature resolution of 1 °C over fiber lengths of up to 50 km and a spatial resolution better than 10 m, but it had no practical application [3]. In 2002, Kirkland* C.J. analyzed the severe fire that occurred in the Channel Tunnel in November 1996, and discussed the importance of fire emergency management and early warning [4]. In 2005,

Kelsey A. et al. studied the detection system composed of combustible gas detectors, but both the accuracy and corresponding time were not ideal [5]. In 2009, Zhan Y. et al. designed a new fiber Bragg grating (FBG) high-temperature sensor capable of achieving a detection range of 0–800 °C and accuracy of ±1 °C, which proves the practical application performance of the fiber optic sensor [6]. In 2015, a fire detection and recognition approach based on the mechanism of visual attention was proposed by Zhang H.J. et al. The flame recognition accuracy reached 82%, unweighted fire comprehensive recognition accuracy reached 76%, and the unweighted omission rate was just 11% [7]. However, this method still belongs to the traditional video detection system, which cannot achieve passive monitoring and requires high maintenance costs. In 2018, Muhammad K. et al. developed an efficient fire detection and localization method based on a deep convolutional neural network (CNN), which can balance the model's size and precision in detecting fire well, but still requires a complete set of video surveillance systems [8]. In 2019, Yan B. et al. proposed a novel Raman-distributed temperature sensor, which can measure temperature with an accuracy of ±1.9 °C at a sensing range of 18.27 km [9]. In 2021, Lv J.D. et al. proposed a technique based on ensemble learning of CNNs for forecasting the unsettling location of the Sagnac-distributed optical fiber sensing system. The mean absolute error of this method does not exceed 14.6 m, and the positioning resolution is 10 m [10]. In 2022, Yang M. and others proposed a composite prediction framework consisting of a convolution neural network and double clustering, which reduced data redundancy and provided better theoretical guidance for the stable and safe operation of the power grid [11]. Given the shortcomings of the above research, this paper proposes an automatic monitoring and early warning system for ultra-weak FBG temperature-sensing of fires based on a composite algorithm combining the distributed individuals differential evolution (DIDE) algorithm and the four-point positioning (FPP) method, i.e., the DIDE-FPP composite algorithm. The DIDE algorithm was proposed by Chen Z.Q. et al. in 2020, and successfully resolves the two difficult issues in multimodal multiobjective optimization problems (MMOPs) of finding more peaks and improving the solution accuracy of the obtained peaks [12].

Based on the above research, we propose the DIDE-FPP composite algorithm by combining it with the four-point positioning (FPP) method. Firstly, the temperature change in the tunnel is monitored by an ultra-weak FBG sensing array composed of a large number of fiber optic temperature sensors and converted into wavelength signals. Then, these wavelength signals are peak-found and positioned by the DIDE algorithm, and after adaptive range adjustment (ARA), lifetime mechanism (LTM), and elite learning mechanism (ELM), the method is optimized and iterated. Lastly, the final found peak point is calculated through the FPP method for three-dimensional geometry to complete the spatial positioning.

## 2. Algorithm and Principle

In this paper, the DIDE-FPP algorithm that we use to locate the tunnel fire is based on the DIDE algorithm combined with the FPP algorithm, with the latter to complement the three-dimensional spatial positioning function of the former. When the ultra-weak FBG fiber grating array is affected by the external temperature, the spectral signal changes, and the peak points with abnormal temperature changes are found through the DIDE. These peak points are then imported into the FPP algorithm for accurate three-dimensional spatial positioning (large-capacity ultra-weak FBG fiber gratings). The position of each sensor of the array is determined at the time when the array was laid. The specific process is shown in Figure 1.

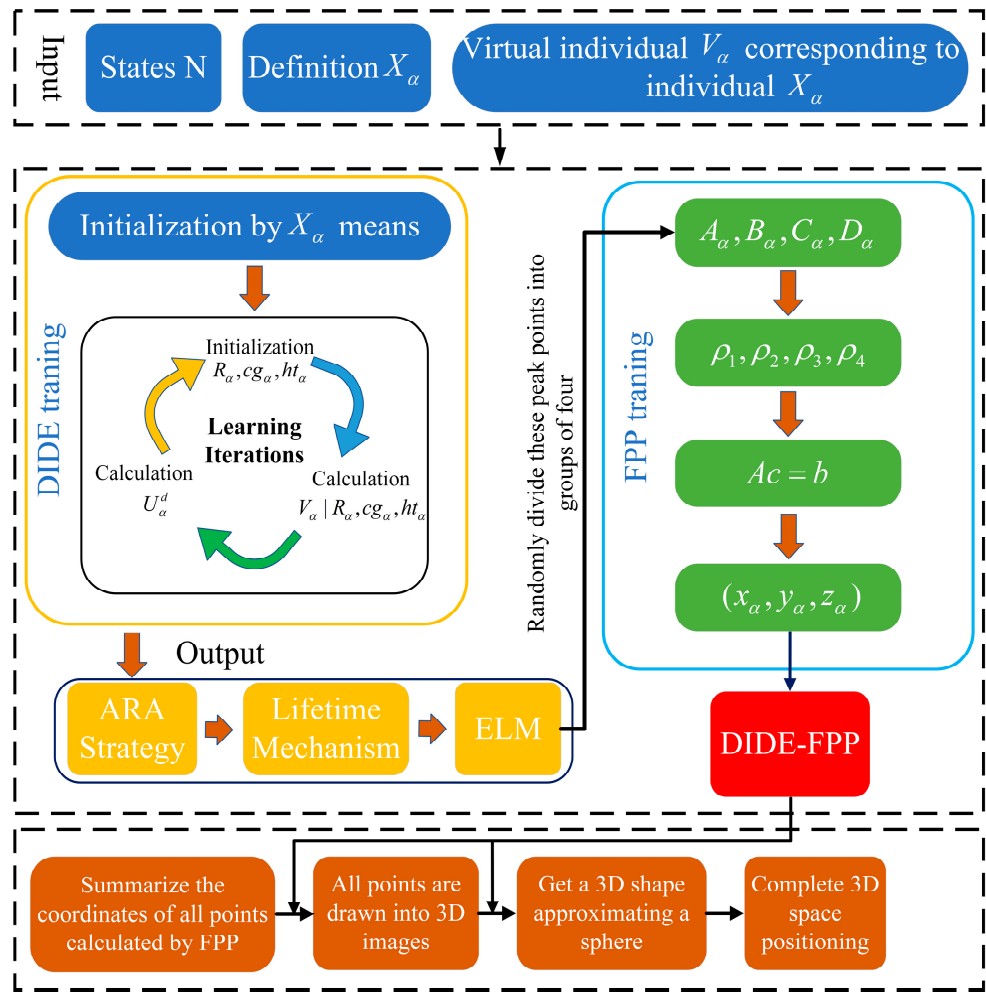

**Figure 1.** Construction and identification of DIDE-FPP.

### 2.1. DIDE Algorithm

A distributed individual's differential evolution is known as DIDE. The distributed individuals for multiple peaks (DIMP) framework, ARA, LTM, and ELM serve as the foundation for the algorithm. Among them, the LTM enhances the population diversity to locate more peaks, and the ELM can improve the peak finding accuracy. The DIDE algorithm consists of the following parts. First, the original signals are numbered as specific individuals $X_\alpha$, $X_\alpha$ and their corresponding $R$, $cg$, $ht$ are initialized; then, two dummy components $VX_{\alpha,1}$, $VX_{\alpha,2}$ are generated for each person $X_\alpha$ as shown in Figure 2, and according to (2), the mutation operation is performed to generate the corresponding $V_\alpha$ ($V_\alpha$'s position will be confined to the associated boundary if it is outside of the search area). After that, a descendant $U_\alpha$ is generated. In the selection process, the generation with the better fitness between $U_\alpha$ and $X_\alpha$ is greedily chosen for the following generation. At the same time, the $X_\alpha$'s cg is updated to assist with ARA strategy. After all individuals finished a generation, ARA, LTM, and ELM are executed sequentially. If the termination requirement is satisfied, DIDE terminates; otherwise, it jumps back to generation steps of $VX_{\alpha,1}$, $VX_{\alpha,2}$ and starts a new build. The above steps are marked in Figure 3.

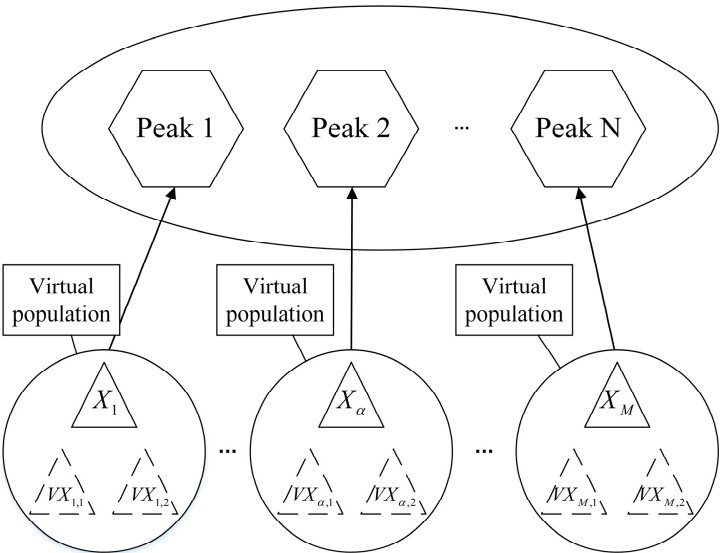

**Figure 2.** The basic framework of DIDE.

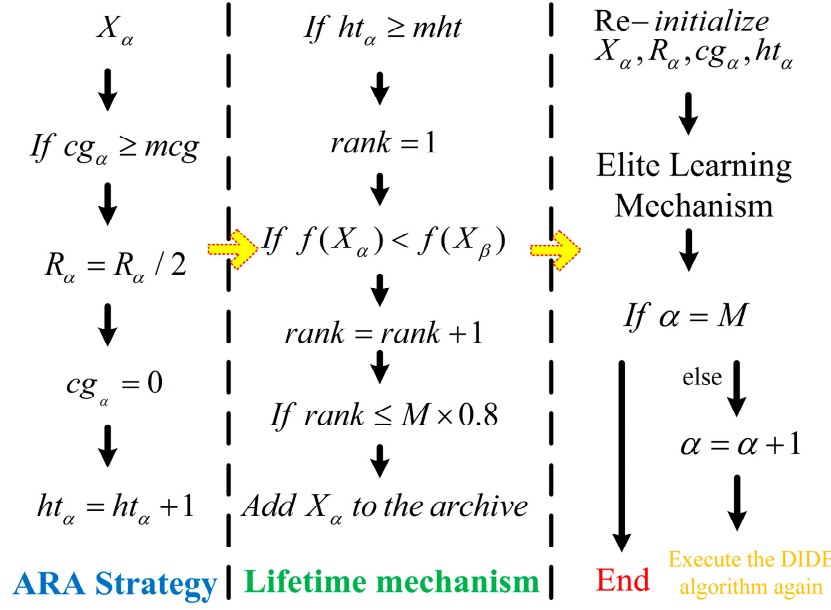

**Figure 3.** The specific implementation process of DIDE.

N is the population size.

$X_\alpha$ is the $\alpha^{th}$ individual in the population.

$VX_{\alpha,1}$ and $VX_{\alpha,2}$ are the virtual individuals in the virtual population of $X_\alpha$.

$$V_\alpha = X_\alpha + F \times (VX_{\alpha,1} - VX_{\alpha,2}) \tag{1}$$

$$VX_\alpha^d = Random\left(\max\left(X_\alpha^d - \frac{R_\alpha^d}{2}, L^d\right), \min\left(X_\alpha^d + \frac{R_\alpha^d}{2}, U^d\right)\right) \tag{2}$$

Note: *F* is a parameter called the scale factor.

Setting the R value is crucial for the DIDE. A higher R improves the peak-finding performance of the algorithm but is less conducive to development, whereas a smaller R does the opposite. Therefore, to balance the peak-finding performance and the development performance, each individual's value of R will need to be identified, and the ARA strategy will need to be used to adaptively alter it. The ARA strategy's operational procedure is depicted in Figure 3. An indicator cg is used in the ARA method to keep track of the

number of iterations during which an individual fails to improve (and is not replaced by virtual offspring during the selection operation). The ARA strategy will halve R to improve development performance whenever an individual's cg value reaches mcg (maximum continuous band), and the associated cg value is then zeroed. At the same time, the ARA strategy is utilized to record the R halvings. Ht (the halving time) will be increased by one each time.

$$U_\alpha^d = \begin{cases} V_\alpha^d, if \ Random(0,1) \leq CR \ or \ d = rnbr_\alpha \\ X_\alpha^d, otherwise \end{cases} \tag{3}$$

$$mcg = 10 \times 2^{[D/10]+1} \tag{4}$$

Inspired by the natural aging phenomenon, a lifecycle mechanism is proposed, which allows the DIDE algorithm to locate more peaks. As shown in Figure 3, if the ht of an individual reaches mht(maximum halving time), the individual will end the cycle (similar to the end of a person's life). If the ht value at the end of the cycle meets the preset criteria for the individual, it is considered as an elite solution and will be included in the designated dossier. The individual's R, cg, and ht will be re-initialized after judgment. This would considerably improve the diversity to find new peaks by preserving the elite data from individual iterations in the archive.

Figure 4 shows the peak-finding simulation results of the DIDE algorithm on the simulation model of the grating array in the tunnel before the iterative optimization, in which each FBG sensor in the array is separated by 10 cm, working in the band of wavelength demodulation of 1550 nm, and the light power is used to indicate the recognition of abnormal temperature detected by the FBG sensor. In the figure, we use different hues to represent the temperature and, the higher the temperature, the warmer the hue gradually becomes.

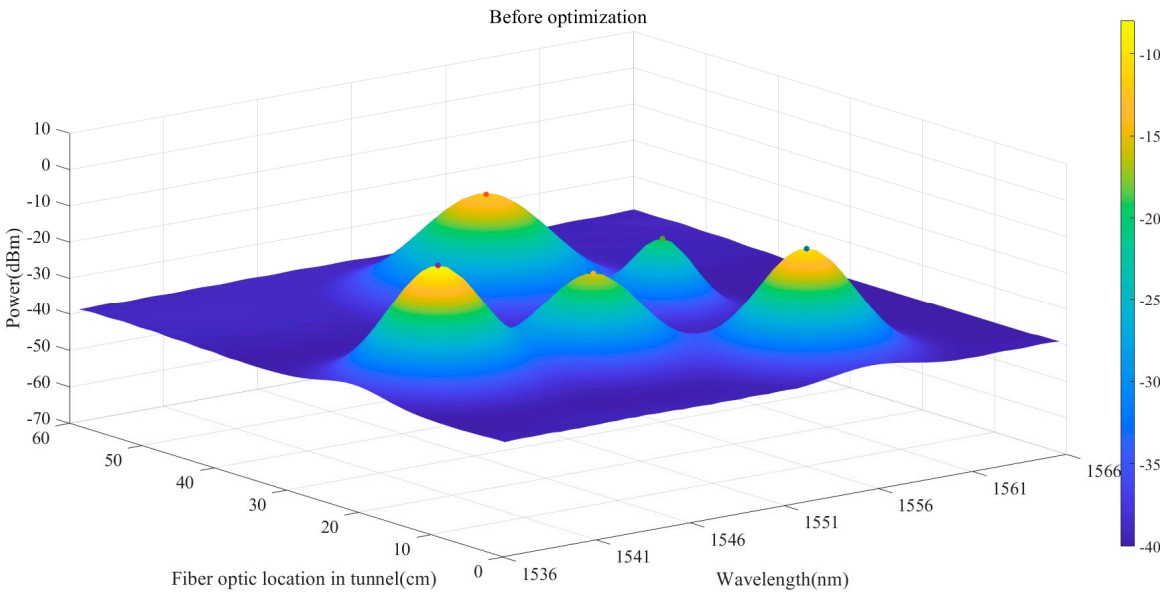

**Figure 4.** DIDE algorithm tunnel simulation environment preliminary peak-finding renderings.

With the continuous iteration of the evolutionary algorithm through the ARA strategy, the life cycle mechanism, and the ELM elite evolution mechanism, after 100 iterations as shown in Figure 5, the number of peaks found by the simulation reaches the theoretical maximum value.

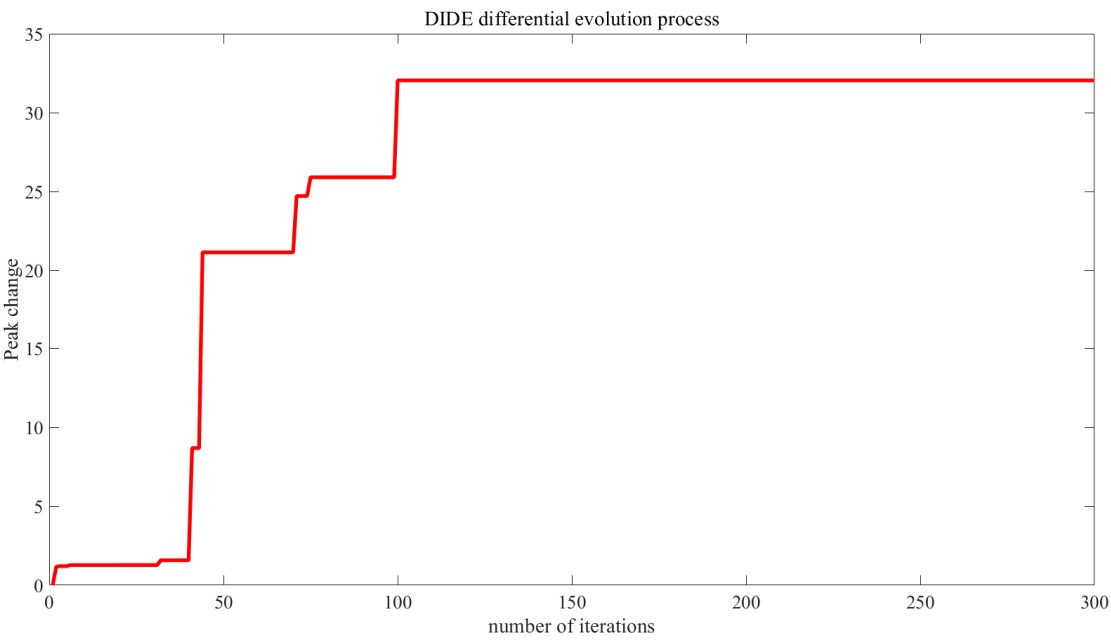

**Figure 5.** The variation in the number of peaks discovered by the DIDE algorithm over time.

From Figure 6, we can see that after all the iterative processes of the DIDE algorithm, the accuracy of the main positioning points can be ensured while identifying more peaks.

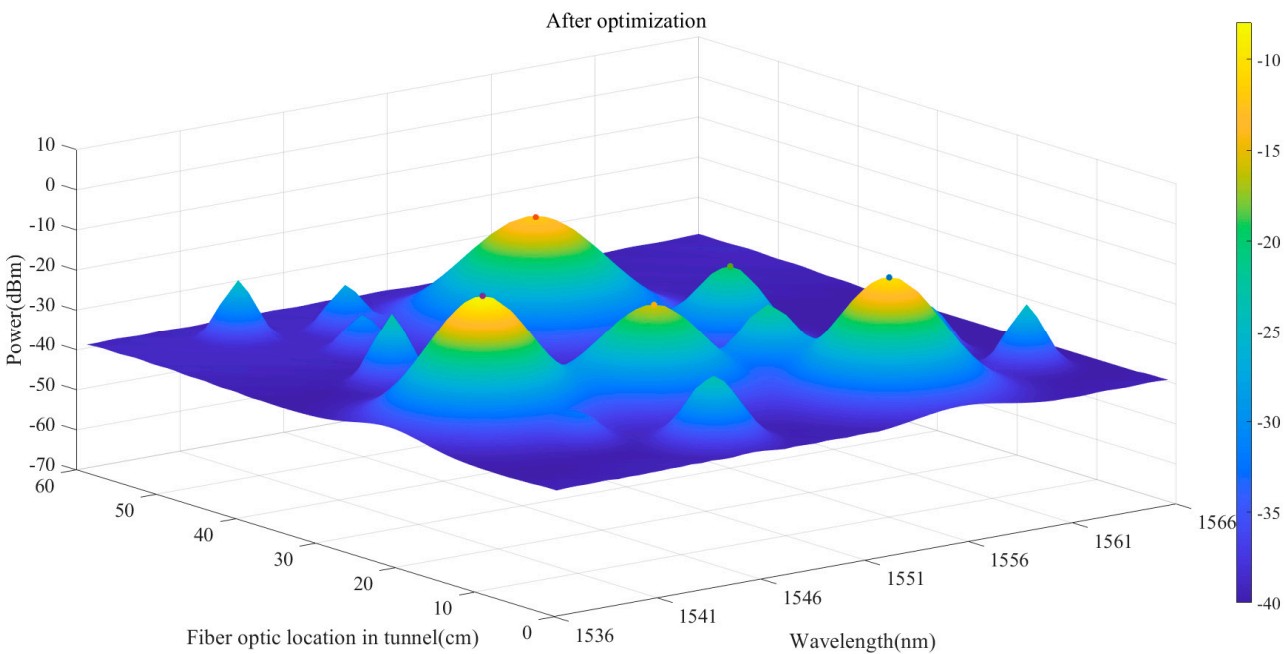

**Figure 6.** Peak-finding renderings after iteration by the DIDE differential evolution algorithm.

### 2.2. FPP Algorithm

The FPP (four-point positioning) algorithm is mostly used for satellite positioning, in which two coordinate points determine a straight line, three coordinate points a plane, and four coordinate points and their distances to the same unknown point determine the coordinates of such a point in the three-dimensional space. Knowing all the four points in space and their distances to any other point, four quadratic equations can be obtained through the distance relationship. After eliminating the higher-order terms through three sets of differences, the equations system is obtained, and the coordinates of any point can be solved using the matrix operations.

Suppose four points A, B, C, and D are known in the space and the coordinates are $(x_1, y_1, z_1)(x_2, y_2, z_2)(x_3, y_3, z_3)(x_4, y_4, z_4)$, suppose we want to calculate the coordinates of an arbitrary point P in the same space, set the distance from point P to A, B, C, and D as $\rho_1$, $\rho_2$, $\rho_3$, and $\rho_4$. According to the distance formula, the following equations can be obtained:

$$
\begin{cases}
(x_1 - x)^2 + (y_1 - y)^2 + (z_1 - z)^2 = \rho_1^2 \\
(x_2 - x)^2 + (y_2 - y)^2 + (z_2 - z)^2 = \rho_2^2 \\
(x_3 - x)^2 + (y_3 - y)^2 + (z_3 - z)^2 = \rho_3^2 \\
(x_4 - x)^2 + (y_4 - y)^2 + (z_4 - z)^2 = \rho_4^2
\end{cases}
\tag{5}
$$

Arrange to obtain:

$$
\begin{cases}
x^2 - 2x_1 x + y^2 - 2y_1 y + z^2 - 2z_1 z = \rho_1^2 - x_1^2 - y_1^2 - z_1^2 \\
x^2 - 2x_2 x + y^2 - 2y_2 y + z^2 - 2z_2 z = \rho_2^2 - x_2^2 - y_2^2 - z_2^2 \\
x^2 - 2x_3 x + y^2 - 2y_3 y + z^2 - 2z_3 z = \rho_3^2 - x_3^2 - y_3^2 - z_3^2 \\
x^2 - 2x_4 x + y^2 - 2y_4 y + z^2 - 2z_4 z = \rho_4^2 - x_4^2 - y_4^2 - z_4^2
\end{cases}
\tag{6}
$$

It can be seen the above formula contains high-order terms, which is not conducive to the solution of the equation. The third-order difference of the equation can eliminate the high-order terms, and the following first-order equation system can be obtained:

$$
\begin{cases}
\{2x(x_2 - x_1) + 2y(y_2 - y_1) + 2z(z_2 - z_1) = \rho_1^2 - \rho_2^2 + x_2^2 - x_1^2 + y_2^2 - y_1^2 + z_2^2 - z_1^2 \\
\{2x(x_3 - x_1) + 2y(y_3 - y_1) + 2z(z_3 - z_1) = \rho_1^2 - \rho_3^2 + x_3^2 - x_1^2 + y_3^2 - y_1^2 + z_3^2 - z_1^2 \\
\{2x(x_4 - x_1) + 2y(y_4 - y_1) + 2z(z_4 - z_1) = \rho_1^2 - \rho_4^2 + x_4^2 - x_1^2 + y_4^2 - y_1^2 + z_4^2 - z_1^2
\end{cases}
\tag{7}
$$

Express the equation in matrix form:

$$
Ac = b \tag{8}
$$

Thereinto:

$$
A = \begin{bmatrix}
2(x_2 - x_1) & 2(y_2 - y_1) & 2(z_2 - z_1) \\
2(x_3 - x_1) & 2(y_3 - y_1) & 2(z_3 - z_1) \\
2(x_4 - x_1) & 2(y_4 - y_1) & 2(z_4 - z_1)
\end{bmatrix}
\tag{9}
$$

$$
c = \begin{pmatrix} x \\ y \\ z \end{pmatrix}
\tag{10}
$$

$$
b = \begin{pmatrix}
\rho_1^2 - \rho_2^2 + x_2^2 - x_1^2 + y_2^2 - y_1^2 + z_2^2 - z_1^2 \\
\rho_1^2 - \rho_3^2 + x_3^2 - x_1^2 + y_3^2 - y_1^2 + z_3^2 - z_1^2 \\
\rho_1^2 - \rho_4^2 + x_4^2 - x_1^2 + y_4^2 - y_1^2 + z_4^2 - z_1^2
\end{pmatrix}
\tag{11}
$$

It can be seen from the operation of the matrix that the coordinates of the point to be located, $(x, y, z)$, are required, that is, the solution vector c needs to be solved. If the inverse matrix of the matrix A exists, the vector c can be solved using the following equation: $c = A^{-1}b$. So far, the coordinates of the point to be solved are solvable. When selecting the known points, the invertibility of the matrix A needs to be ensured. From a geometric point of view, it is necessary to ensure that the four known points are not in the same plane. Figure 7 shows the single positioning effect of FPP algorithm.

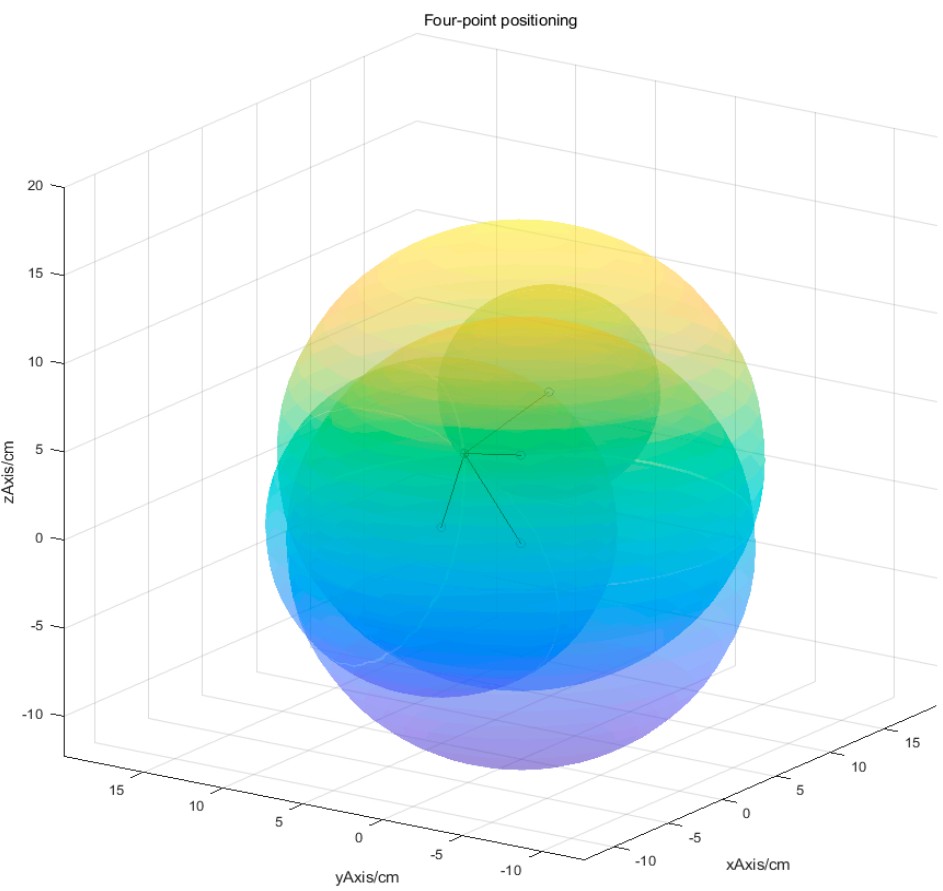

**Figure 7.** FPP algorithm running demo.

### 3. Experimental Comparison

We employ peak ratio (PR) and success rate (SR) for the performance evaluation of these functions in order to impartially assess the peak-finding performance of the DIDE-FPP composite algorithm. Given the maximum number of finite elements (MaxFEs) and the accuracy $\varepsilon$, the expressions of PR and SR are as follows:

$$PR = \frac{\sum\limits_{\alpha=1}^{NR} NPF_\alpha}{TNP \times NR} \qquad (12)$$

$$SR = \frac{NSR}{NR} \qquad (13)$$

where PR is the average percentage of global peaks discovered across many runs, NR stands for number of runs, $NPF_\alpha$ is the quantity of worldwide peaks discovered in the $\alpha th$, and the number of global peaks in the optimization problem is referred to as TNP; SR is the proportion of successful runs among many runs, and NSR is the quantity of successful runs among all NR runs (successful runs are those that locate every global peak). More specifics on these two efficiency indicators can be found in [12].

To assess the effectiveness of the DIDE-FPP algorithm in peak detection and solving MMOPs, we used 20 multimodal test functions. This is also the CEC'2013 benchmark set that is widely used in evaluating algorithm peak-finding performance and solving MMOP problems [13]. The following Table 1 displays the fundamental characteristics of these 20 functions.

**Table 1.** Details of the 20 functions used.

| Function | Name | Local Optimum | Global Optimum | Dimesion Size |
|---|---|---|---|---|
| $F_1$ | 0 | 3 | 2 | 1 |
| $F_2$ | Equal Maxima | 0 | 5 | 1 |
| $F_3$ | Uneven Decreasing Maxima | 4 | 1 | 1 |
| $F_4$ | Himmelblau | 0 | 4 | 2 |
| $F_5$ | Six-Hump Camel Back | 2 | 2 | 2 |
| $F_6$ | Shubert with 2D | many | 18 | 2 |
| $F_7$ | Vincent with 2D | 0 | 36 | 2 |
| $F_8$ | Shubert with 3D | many | 81 | 3 |
| $F_9$ | Vincent with 3D | 0 | 216 | 3 |
| $F_{10}$ | Modified Rastrigin | 0 | 12 | 2 |
| $F_{11}$ | Composition Function 1 with 2D | many | 6 | 2 |
| $F_{12}$ | Composition Function 2 with 2D | many | 8 | 2 |
| $F_{13}$ | Composition Function 3 with 2D | many | 6 | 2 |
| $F_{14}$ | Composition Function 3 with 3D | many | 6 | 3 |
| $F_{15}$ | Composition Function 4 with 3D | many | 8 | 3 |
| $F_{16}$ | Composition Function 3 with 5D | many | 6 | 5 |
| $F_{17}$ | Composition Function 4 with 5D | many | 8 | 5 |
| $F_{18}$ | Composition Function 3 with 10D | many | 6 | 10 |
| $F_{19}$ | Composition Function 4 with 10D | many | 8 | 10 |
| $F_{20}$ | Composition Function 4 with 20D | many | 8 | 20 |

$F_1$, $F_2$, and $F_3$ are basic functions in one dimension. $F_4$ and $F_5$ are non-extendable two-dimensional functions. $F_6$–$F_{10}$ are extendable 2D or 3D functions. $F_{11}$–$F_{20}$ are intricate composite functions created from a number of fundamental functions with various features. These 20 functions are divided into low-dimensional ($F_1$–$F_{17}$) functions in 1D or 5D, and high-dimensional ($F_{18}$–$F_{20}$) functions in 10D or 20D.

This study compares 13 cutting-edge multimodal optimization techniques with our proposed DIDE-FPP algorithm, including CDE [14], SDE [15], NCDE, NSDE [16], MOM-MOP [17], LIPS [18], NMMSO [19], LoICDE, LoISDE [20], PNPCDE [21], EMO-MMO [22], self-CCDE, and self-CSDE [23]. Except for MOMMOP, NMMSO, and EMOMMO, which have unique population size setting strategies to deal with the CEC'2013 benchmark set, all of the compared algorithms and our proposed DIDE-FPP algorithm have a population size of 100. Other parameters in the comparison method are configured based on the references for those parameters.

The F and crossover rate (CR) are set to 0.3 and 0.9 in DIDE-FPP. The mht in the LTM is 10. The drop threshold (dt) in ELM is 40, whereas $\sigma_\beta$ and $\sigma_{\min}$ are, respectively, $1.0 \times 10^{-4}$ and $1.0 \times 10^{-10}$.

The DIDE-FPP method is compared to the other 13 algorithms on the function $F_1$−$F_{20}$ in Figure 8, using the PR value as the evaluation index. The precision level used in the experiment is $\varepsilon = 1.0 \times 10^{-4}$, and Table 2 displays the MaxFEs parameters for various functions. For a fair comparison, the average outcomes of 50 independent runs were utilized.

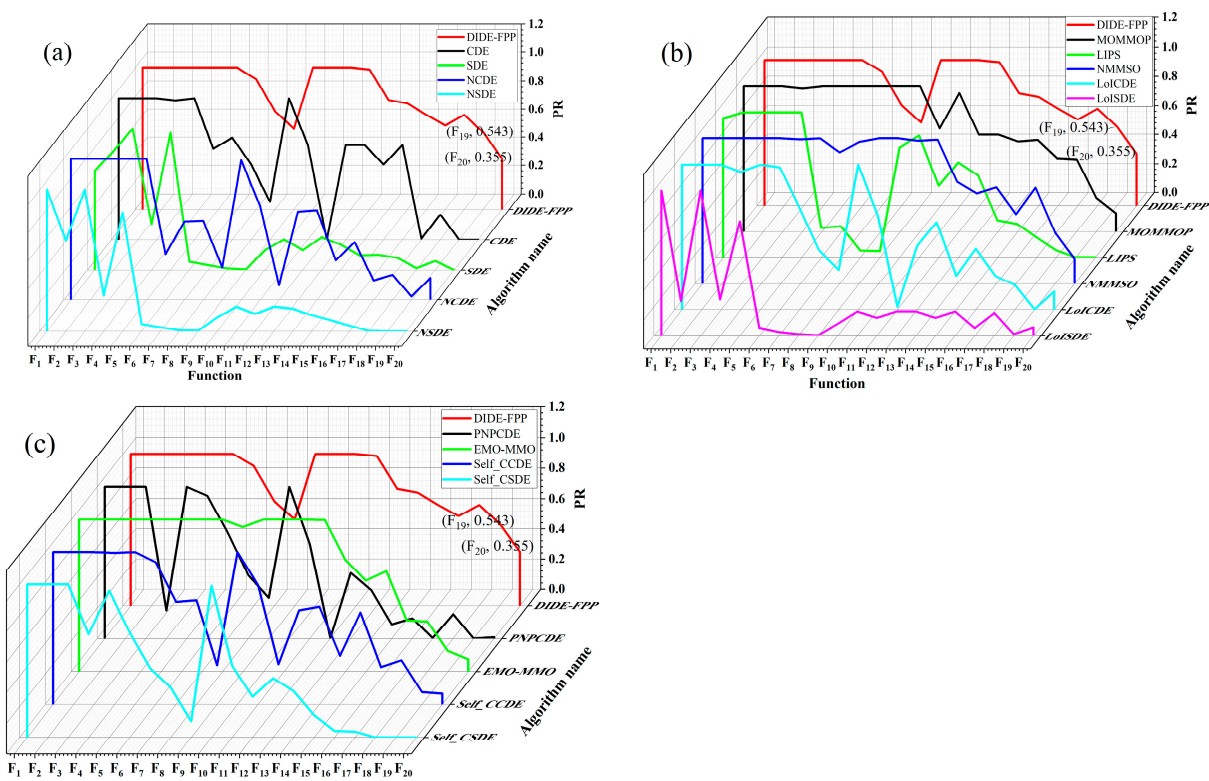

**Figure 8.** Comparison of PR values between the DIDE-FPP algorithm and other algorithms. (**a**–**c**) The 13 algorithms are divided into three groups for comparison.

**Table 2.** Setting of MaxFEs.

| Function | MaxFEs |
|---|---|
| $F_1$−$F_5$ | $5.0 \times 10^4$ |
| $F_6$−$F_7$ | $2.0 \times 10^5$ |
| $F_8$−$F_9$ | $4.0 \times 10^5$ |
| $F_{10}$−$F_{13}$ | $2.0 \times 10^5$ |
| $F_{14}$−$F_{20}$ | $4.0 \times 10^5$ |

In Figure 9, DIDE-FPP is marked with red lines. We can easily see from the three-dimensional diagram that among all 20 functions, DIDE-FPP has the best average performance, and waterfall accounts for the highest proportion of all tested algorithms. The area is also the largest.

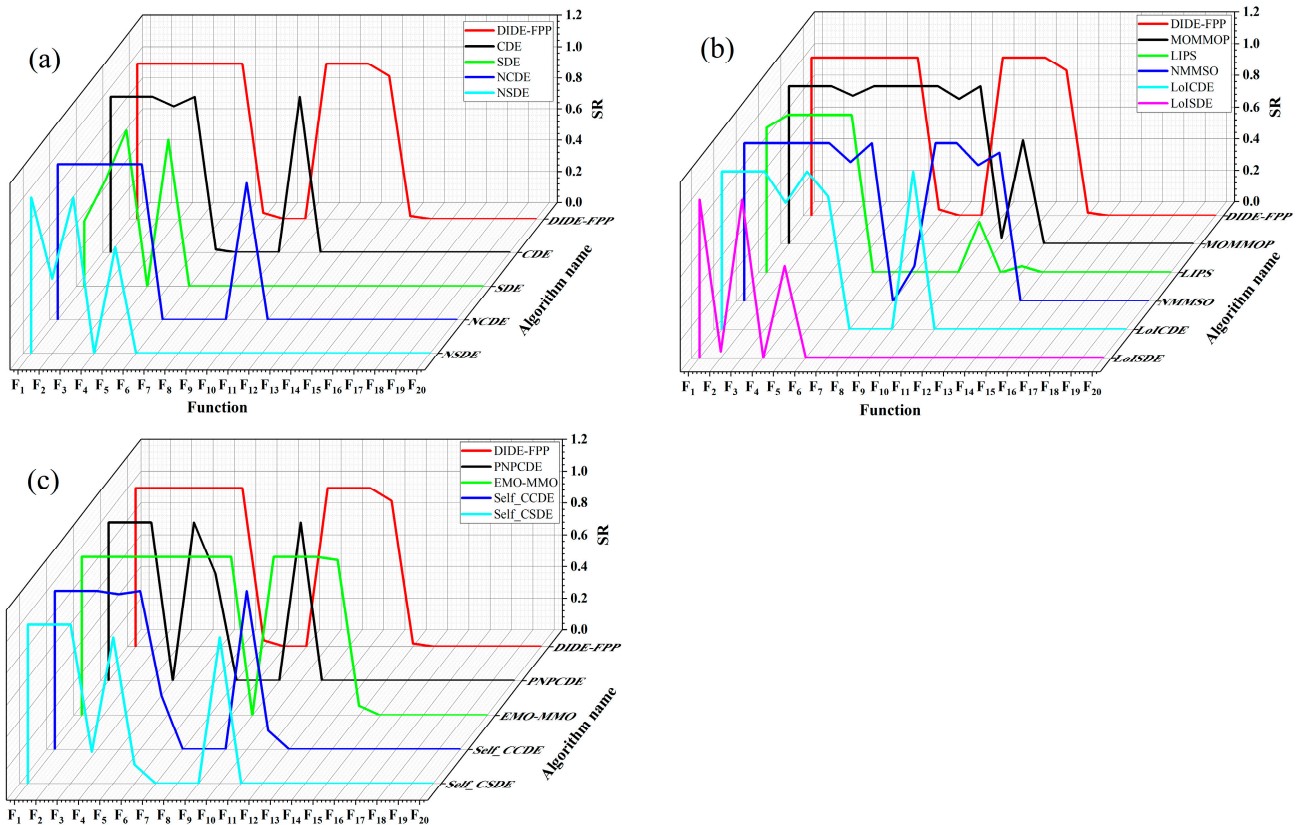

**Figure 9.** Comparison of SR values between the DIDE-FPP algorithm and other algorithms. (**a**–**c**) divides 13 algorithms into three groups for comparison.

DIDE-FPP identifies all peaks on $F_1$–$F_6$ and $F_{10}$ in a stable manner (i.e., an SR result of 1.000). Only EMO-MMO can match the performance of DIDE-FPP among all the algorithms that were examined. While NMMSO fails at $F_6$, MOMMOP fails at $F_4$. On at least two functions, other approaches are ineffective. On $F_7$–$F_9$, MOMMOP, NMMSO, and EMO-MMO exceed DIDE-FPP and attain extremely good performance. The result of DIDE-FPP on $F_7$–$F_9$, however, is still incredibly encouraging and greatly outperforms the other ten algorithms that were examined.

The complex combinatorial functions $F_{11}$–$F_{20}$ are an area where DIDEFPP outperforms other competing algorithms. DIDE-FPP identifies all peaks stably on $F_{11}$–$F_{12}$. On $F_{13}$, despite the fact that EMO-MMO has the best performance, the Wilcoxon rank-sum test finds no significant difference in performance between DIDE-FPP and EMO-MMO. DIDE-FPP outperforms all other examined algorithms on $F_{14}$–$F_{20}$. EMO-MMO performs effectively between $F_{11}$ and $F_{13}$ but fails between $F_{14}$ and $F_{20}$. DIDE-FPP beats EMO-MMO in six functions, $F_{14}$, $F_{15}$, and $F_{17}$–$F_{20}$.

In particular, DIDE-FPP significantly outperforms other algorithms on $F_{19}$–$F_{20}$ (two 10D and 20D high-dimensional functions, respectively). Due to their high dimensional size, compared to other combined functions, these two are more challenging. Very few peaks can be found by many algorithms at $F_{19}$ and $F_{20}$, or they find none at all, namely CDE, SDE, NSDE, LIPS, LoISDE, PNPCDE, Self CCDE, and Self-CSDE. Furthermore, only a very small number of peaks can be found with NCDE and LoICDE on $F_{19}$, and the same is true for EMO-MMO on $F_{20}$. DIDE-FPP's PR outcomes on $F_{19}$ and $F_{20}$ are 0.543 and 0.355,

respectively, while among all the compared algorithms on these two functions, the best PR results are only 0.350 and 0.170.

Here, we count the data comparison results (PR, SR) of DIDE-FPP and 13 other algorithms on 20 functions. In Figure 10, we divide the comparison results between DIDE-FPP and 13 algorithms into there categories.

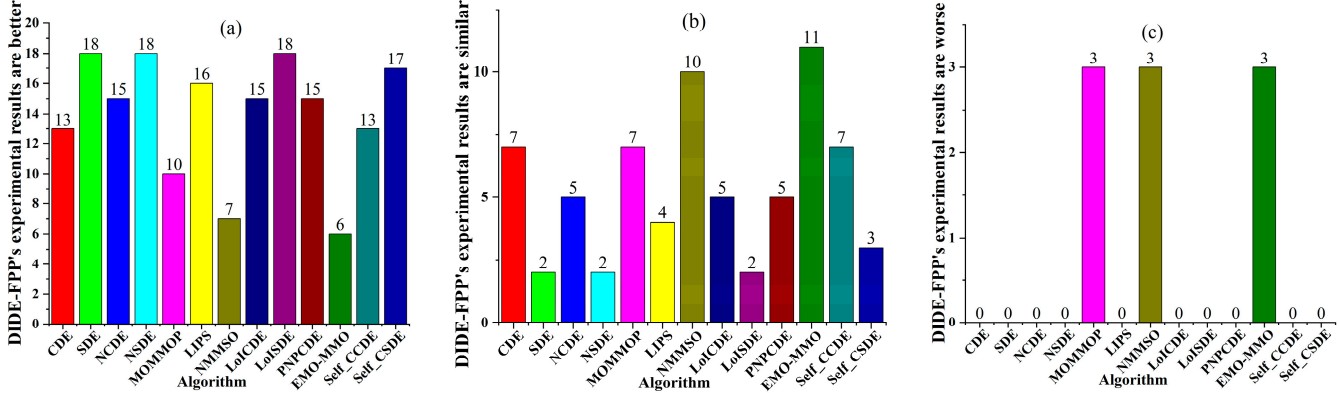

**Figure 10.** A comprehensive reference comparison summary of PR and SR results of A and 13 algorithms, where (**a**) is those with better results in 20 functions, (**b**) is those with similar results in 20 functions, and (**c**) is those with worse results in 20 functions.

When compared to either algorithm, the number of functions where DIDE-FPP performs considerably better is more than the number of functions where DIDE-FPP performs significantly worse, showing that DIDE-FPP performs better overall. For the experimental results of DIDE and other 13 comparison algorithms in other accuracy levels such as $\varepsilon = 1.0 \times 10^{-5}$, please refer to [13].

## 4. Simulation and Discussion

### 4.1. Experimental Scenario

The experimental area is the Changchun Liaoning Road station metro tunnel, and each FBG array has a total length of about 10 km and a spatial resolution of 100 mm. The system laying site is shown in Figure 11a, a 10 km long subway tunnel. The optical cable used by the monitoring system is shown in Figure 11b, and its length is 10 km, as long as the tunnel. The distance between each optical fiber is 10 cm, as shown in Figure 11c. Figure 11d shows the actual operation state of the metro tunnel after the laying of the optical fiber array.

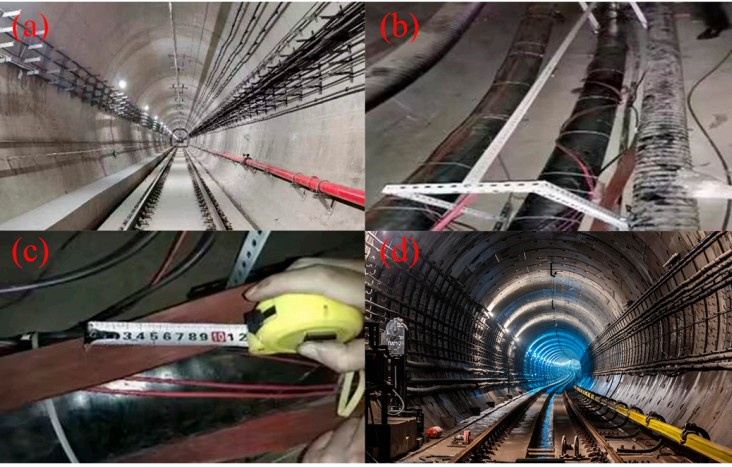

**Figure 11.** Subway tunnel monitoring system. (**a**) Metro tunnel. (**b**) Optical cable. (**c**) Optical fiber distance. (**d**) Tunnel laying completed.

### 4.2. Simulation System

The specific structure of an FGB tunnel fire online monitoring and early warning system based on the DIDE-FPP composite algorithm proposed in this paper is shown in Figure 12. The optical signal is emitted by the ASE (amplified spontaneous emission) light source. After entering the FBG sensing array through the optical coupler, each FBG sensor in the FBG sensing array will reflect its different wavelength signal into the wavelength demodulation system through the optical circulator again. Because the central wavelength of each sensor does not overlap, the demodulation system will demodulate the returned single wavelength signal one by one, which corresponds to the fixed measured reference point in the FBG sensing array, and the wavelength signal is the sensing information of this measurement point. At this time, the DIDE-FPP algorithm is used to find the accurate FBG sensor corresponding to the wavelength signal and to determine the location of the temperature anomaly in the tunnel. Firstly, through a series of evolutionary iterations by the DIDE algorithm, the peaks are successfully found, and the FBG sensors affected by high temperature(>40 °C) are determined (the corresponding virtual population is generated according to each sensor individual, and then iterated and filtered through ARA strategy, LTM and ELM in turn). Then, these discrete points (several FBG sensors) locate the specific area with the abnormal temperature, and upload the data to the upper computer and alarm system after completing the fire positioning, which achieves the early warning and real-time monitoring of the fire.

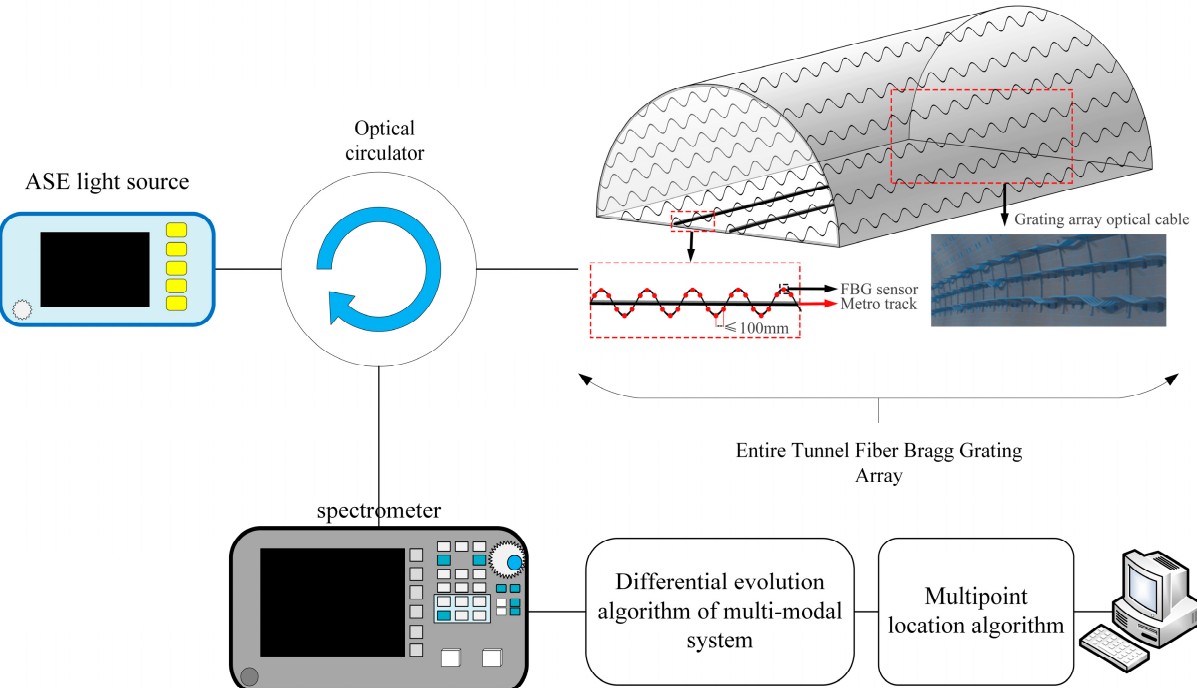

**Figure 12.** Structure diagram of tunnel fire location system.

The grating array used in the system consists of a large number of ultra-weak FBGs. Usually, we call fiber grating with reflectivity less than −30 dB ultra-weak fiber grating. As shown in Figure 13, the power loss of ultra-weak FBG reflected light is extremely small. After $2 \times 843$ fiber grating transmission attenuations, the reflected light power of FBG843 can still reach about 80% of that of FBG1 [24]. Ultra-weak fiber grating sensor arrays can multiplex a large number of sensors.

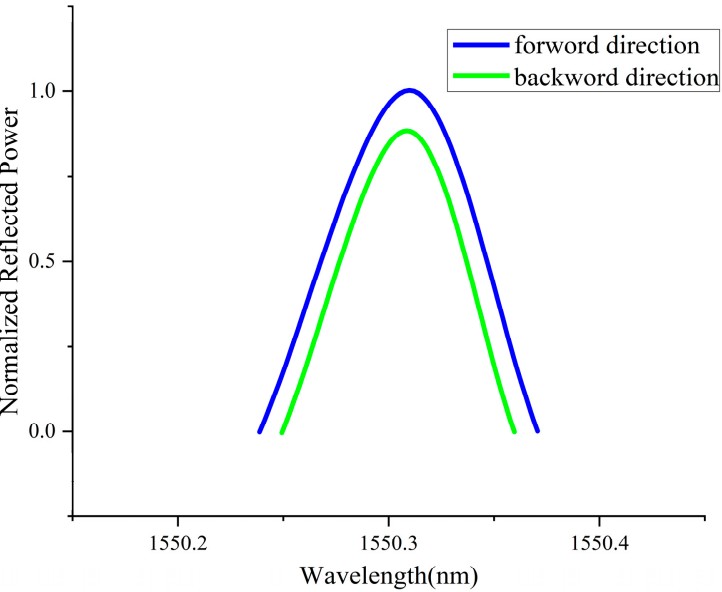

**Figure 13.** Reflected signal of FBG1/FBG843 (reflectivity = −40 dB ~ −45 dB).

In the simulation model, we assume that each FBG sensor in this composite fiber grating array installed in the tunnel is separated by 10 cm. When in the experimental application, the density of the grating can be selected according to the specific situation. The precision of the higher density grating array will increase, but the relative cost increases as well. When fabricating the fiber grating array, the position of each sensor on the fiber is fixed. When the FBG array is installed, the vertical distance $\rho$ of each sensor from the tunnel floor is determined.

As shown in Figure 14, the distance $\rho_\alpha$ of each FBG sensor from the tunnel floor is fixed after the entire tunnel fire monitoring system is laid. When a fire source or a high-temperature abnormal point appears, we can accurately locate the FBG sensors affected by the high temperature (>40 °C) points through the DIDE algorithm. Then, through the FPP method, these identified FBG sensors are grouped into groups of four for three-dimensional spatial positioning multiple times. Finally, a sphere region is identified with its center as the fire source or the abnormal point with higher temperature. Through $c = A^{-1}b$, we can obtain the coordinates of the center of the sphere (the target point of the four-point positioning) region to complete the entire three-dimensional spatial positioning process.

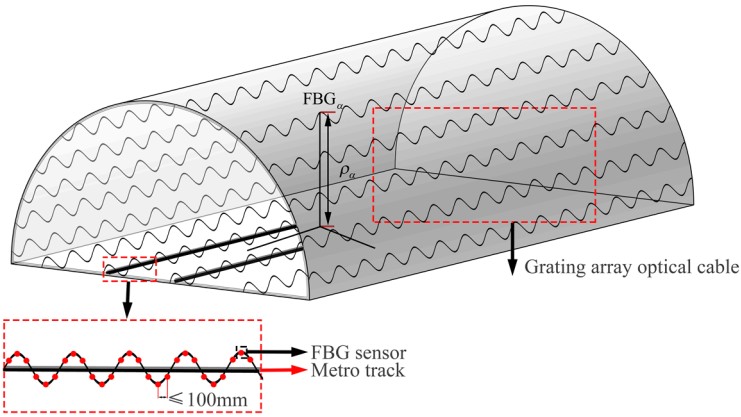

**Figure 14.** An installed FBG sensor and its distance from the ground.

For the determination standard of fire in the monitoring system, refer to the Code for Design of Automatic Fire Alarm System GB 50116-2013 [25]. The system defines the temperature range of 20–40 °C as the normal value. In the monitoring of the DIDE-FPP

peak-finding location method, the change in optical fiber wavelength in its range is defined as a noise signal. The location of these temperatures is not spatially located through iterative peak-finding optimization, so as not to trigger the alarm mechanism. When the system detects that the temperature at any location in the subway tunnel exceeds 40 °C, the specific location of the temperature anomaly point in the subway tunnel is determined through the real-time monitoring of the change in the reflected light wavelength of the optical fiber sensor installed at these locations and the DIDE-FPP spatial positioning method. The detailed workflow is shown in Figure 15 below.

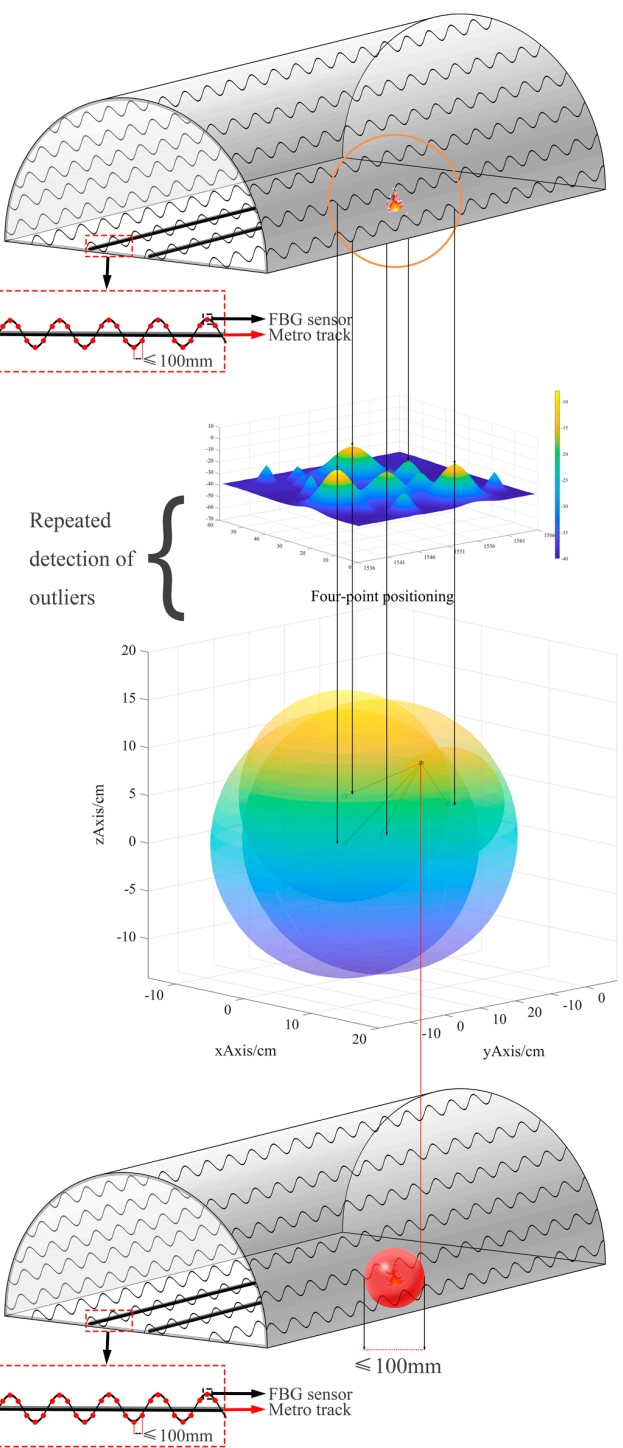

**Figure 15.** Simulation of the location of the ignition point.

The larger outer orange circle in Figure 15 represents the range of all FBG sensors affected by the ignition point. With the temperature decreasing gradually from the center point to the edge, the FBG sensors are less and less affected. After the C-band optical signal emitted by the ASE light source enters the FBG sensor array through the optical coupler, all FBG sensors in the FBG array will reflect their received signals into the wavelength demodulation system, which then pass through the optical circulator. The wavelengths reflected by these FBG sensors affected by abnormal temperature will first be identified by the DIDE peak-finding algorithm, and then positioned in the three-dimensional space by the FPP algorithm. The positioning interval of the DIDE-FPP algorithm is indicated by the red sphere in Figure 15.

Since the C-band signal emitted by the ASE light source of the system is a simple function, even if multiple fibers are used at the same time, the peak-finding operation of the non-complex function is repeated. The peak-finding recognition accuracy of the DIDE algorithm under a simple function is almost 100%, meaning that the average indices of the functions $F_1–F_6$, PR and SR, are one. Finally, the 3D spatial positioning is performed by the FPP (in principle, four-point spatial positioning will not lose accuracy).

Due to the special requirements for monitoring small-sized and weak fire sources in subway tunnels, it is necessary for us to use the ultra-weak fiber grating sensor arrays in practical applications. The multiplexing capacity of the ultra-weak fiber grating sensor array has been greatly improved, and we can determine the distribution density of the FBG sensors in the grating array according to the actual situation. The spatial resolution of the grating can be determined by adjusting the writing pitch during fabrication. Due to the powerful multiplexing ability of ultra-weak FBG gratings, the final spatial positioning accuracy can easily reach the centimeter level.

### 4.3. Discussion

4.3.1. Algorithm Performance Analysis

In order to more intuitively explain the peak finding performance of the DIDE-FPP composite algorithm, we run the five peaking methods DIDE-FPP, MOMMOP, NMMSO, PNPCDE, and Self-CCDE on the high-dimensional function $F_{14}$ in Table 1 50 times, take the average of the results to calculate PR and SR, and then calculate the performance multiplier (PM) through PR and SR.

$$PM = \frac{PR_{other} + SR_{other}}{PR_{DIDE-FPP} + SR_{DIDE-FPP}} \tag{14}$$

Table 3 shows the results of our peak-finding test on the high-dimensional function $F_{14}$. The average PR value of the DIDE-FPP composite method proposed by us is 0.773, and the average SR value is 0.020. Compared with the traditional algorithm, the peak ratio and success rate are significantly improved. Compared with the PNPCDE method, this method greatly improves the comprehensive performance of peak finding. Therefore, this method can realize the accurate location of subway tunnel fires and can be extended to other large fiber grating sensor networks with high accuracy and real-time monitoring.

**Table 3.** Comparison of different peak-finding algorithms.

| Method | PR | SR | PM | Reference |
|---|---|---|---|---|
| MOMMOP | 0.667 | 0.000 | 84.11% | [17] |
| NMMSO | 0.700 | 0.000 | 88.27% | [19] |
| PNPCDE | 0.317 | 0.000 | 39.97% | [21] |
| Self-CCDE | 0.640 | 0.000 | 80.71% | [23] |
| DIDE-FPP | 0.773 | 0.020 | 1 | This work |

### 4.3.2. Analysis of Tunnel Fire Monitoring Technology

Table 4 shows the specific performance analysis results of tunnel fire monitoring technology based on the DIDE-FPP method and three other monitoring technologies. The tunnel fire monitoring technology based on the DIDE-FPP method proposed by us can achieve not only passive monitoring but also centimeter-level positioning accuracy and fast response speed. Compared with other monitoring technologies, under the premise of passive monitoring, the positioning accuracy and response speed also reach an advanced level. Therefore, this technology can realize the accurate location of subway tunnel fires.

**Table 4.** Comparison of technologies available for tunnel fire monitoring.

| Method Name | Accuracy | Power Demand | Response Time | Reference |
|---|---|---|---|---|
| Based on Visual Attention Mechanism | Limited by the accuracy of camera | Power supply required | Slow | [7] |
| Efficient deep CNN-based | Limited by the accuracy of camera | Power supply required | Fast | [8] |
| Based on CNNs ensemble learning | Meter-level | Passive | Fast | [10] |
| Based on DIDE-FPP | Centimeter-level | Passive | Fast | This work |

### 5. Conclusions

(1) The DIDE-FPP composite algorithm proposed in this paper can adaptively process the reflected signals generated in the tunnel fire positioning system and accurately locate the peaks. Compared with 13 other existing advanced algorithms such as CDE and SDE, etc., the comprehensive performance of the proposed DIDE-FPP composite algorithm has improved to some extent, and the combined value of PR and SR on up to 20 experimental functions is relatively high.

(2) The ultra-weak FBG temperature-sensing fire automatic monitoring and early warning system proposed in this paper can realize visual positioning of fires in three-dimensional space, and its spatial positioning accuracy can reach the centimeter level.

**Author Contributions:** Conceptualization, H.J. and Y.Z.; methodology, Y.Z.; software, Y.Z.; validation, H.J., Y.Z. and C.W.; formal analysis, L.C.; investigation, L.C.; data curation, C.W.; writing—original draft preparation, Y.Z.; writing—review and editing, H.J.; project administration, H.J. All authors have read and agreed to the published version of the manuscript.

**Funding:** This research was funded by the China Jilin Provincial Development and Reform Commission Innovation Capacity Building Project, grant numbers 2022C045-1; the Natural Science Foundation of Jilin Province, China, grant number 20210101479JC; and Jilin Provincial Science and Technology Department Project, grant number 20220203051SF.

**Institutional Review Board Statement:** Not applicable.

**Informed Consent Statement:** Not applicable.

**Data Availability Statement:** Not applicable.

**Conflicts of Interest:** The authors declare no conflict of interest.

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
