# Peer review of "A DIDE-FPP Composite Algorithm for Accurate Tunnel Fire Location"

_photonics, doi:10.3390/photonics10030272_

Round 1

Reviewer 1 Report

This present work reports that a DIDE-FPP algorithm can improve the spatial location accuracy of tunnel fire. This article verifies the advantages of DIDE-FPP algorithm by comparing its performance with other 13 algorithms. Before the acceptance, the authors should further modify the manuscript and response to the following comments.

1. Is "DIDEFPP" in line 278 the same method as DIDE-FPP?

2. The experiment in the article is to simulate and locate the fire in the tunnel, but I haven't found the specific temperature-monitoring range. How to define the standard? How can it be considered as a fire? What exactly does the "high temperature" in the text mean? Please specify.

3. In Figure 2, please make sure the numbers and letters can be seen by the readers.

4. Figure 8 is a table but not a figure. In abstract, MMOPs is not explained first.

5. Please double check the typo in the whole paper.

Reviewer 2 Report

In the paper, a new DIDE-FPP composite method is proposed. This method improves the precision of peak finding and location through the iteration of virtual population. The paper proves the superiority of the DIDE-FPP method through PR and SR evaluation indicators. This new composite method is applied to a tunnel fire monitoring system, and its spatial positioning accuracy can reach 10 cm. The DIDE-FPP algorithm proposed in this article has excellent performance, and its description and performance analysis are very detailed. However, there are several errors that need to be corrected, such as:

1)      The introduction should add more novel achievements (such as the related achievements published in 2022) to enhance the persuasiveness of the advantages of the proposed DIDE-FPP method and tunnel fire monitoring system.

2)      What is the fire judgment standard of the fire monitoring system? The specific standards are not mentioned in the text and should be explained.

3)      Figure 16 describes the work flow of the whole tunnel fire monitoring system, but there is no comparison with other systems to illustrate the advantages of this system. Conventional methods such as video monitoring and infrared hybrid monitoring may also achieve similar results. Therefore, the author is suggested to analyze and compare the working environment, monitoring reliability and accuracy of different systems.

I suggest accepting this document with only minor changes.
